# Evaluation of Strength and Muscle Activation Indicators in Sticking Point Region of National-Level Paralympic Powerlifting Athletes

**DOI:** 10.3390/jfmk6020043

**Published:** 2021-05-15

**Authors:** Felipe J. Aidar, Filipe Manuel Clemente, Dihogo Gama de Matos, Anderson Carlos Marçal, Raphael Fabrício de Souza, Osvaldo Costa Moreira, Paulo Francisco de Almeida-Neto, José Vilaça-Alves, Nuno Domingos Garrido, Jymmys Lopes dos Santos, Ian Jeffreys, Frederico Ribeiro Neto, Victor Machado Reis, Breno Guilherme de Araújo Tinoco Cabral, Thomas Rosemann, Beat Knechtle

**Affiliations:** 1Department of Physical Education, Federal University of Sergipe (UFS), São Cristovão 49100-000, Sergipe, Brazil; fjaidar@gmail.com (F.J.A.); raphaelctba20@hotmail.com (R.F.d.S.); 2Group of Studies and Research of Performance, Sport, Health and Paralympic Sports-GEPEPS, The Federal University of Sergipe (UFS), São Cristovão 49100-000, Sergipe, Brazil; jymmyslopes@yahoo.com.br; 3Graduate Program of Physiological Science, Federal University of Sergipe (UFS), São Cristovão 49100-000, Sergipe, Brazil; dihogogmc@hotmail.com; 4Graduate Program of Physical Education, Federal University of Sergipe (UFS), São Cristovão 49100-000, Sergipe, Brazil; acmarcal@yahoo.com.br; 5Escola Superior Desporto e Lazer, Instituto Politécnico de Viana do Castelo, Rua Escola Industrial e Comercial de Nun’Álvares, 4900-347 Viana do Castelo, Portugal; filipe.clemente5@gmail.com; 6Instituto de Telecomunicações, Delegação da Covilhã, 1049-001 Lisboa, Portugal; 7Cardiovascular & Physiology of Exercise Laboratory, University of Manitoba, Winnipeg, MB R3T 2N2, Canada; 8Campus of Florestal, University Federal of Viçosa (UFV), Florestal 36570-900, Minas Gerais, Brazil; ocostamoreira@gmail.com (O.C.M.); brenotcabral@gmail.com (B.G.d.A.T.C.); 9Department of Physical Education, Federal University of Rio Grande do Norte, Natal 59078-970, Rio Grande do Norte, Brazil; paulo220911@hotmail.com (P.F.d.A.-N.); victormachadoreis@gmail.com (V.M.R.); 10Research Center in Sports Sciences, Health Sciences and Human Development (CIDESD), Trás os Montes and Alto Douro University, 5001-801 Vila Real, Portugal; josevilaca@utad.pt (J.V.-A.); ndgarrido@gmail.com (N.D.G.); 11Faculty of Life Sciences and Education, University of South Wales, Pontypridd, Wales CF37 5UP, UK; ian.jeffreys@allproperformance.co.uk; 12Paralympic Sports Program, SARAH Rehabilitation Hospital Network, Brasilia 71535-005, Distrito Federal, Brazil; fredribeironeto@sarah.br; 13Institute of Primary Care, University of Zurich, 8091 Zurich, Switzerland; thomas.rosemann@usz.ch; 14Medbase St. Gallen Am Vadianplatz, 9001 St. Gallen, Switzerland

**Keywords:** strength indicators, sticking point, Powerlifting Paralympic

## Abstract

Background: The sticking region is considered an intervening factor in the performance of the bench press with high loads. Objective: To evaluate the strength indicators in the sticking point region in Powerlifting Paralympic athletes. Methods: Twelve Brazilian Powerlifting Paralympic athletes performed maximum isometric force (MIF), rate of force development (RFD), time at MIF, velocity, dynamic time in sticking, and surface electromyography in several distances from the bar to the chest. Results: For velocity, there was a difference between the pre-sticking and sticking region (1.98 ± 0.32 and 1.30 ± 0.43, *p* = 0.039) and dynamic time between the pre-sticking and the sticking region (0.40 ± 0.16 and 0.97 ± 0.37, *p* = 00.021). In static test for the MIF, differences were found between 5.0 cm and 15.0 cm (CI 95% 784; 1088; *p* = 0.010) and between 10.0 cm and 5.0 cm (CI 95% 527; 768; *p* < 0.001). Regarding the RFD, differences were found (CI 95% 938; 1240; *p* = 0.004) between 5.0 cm and 25.0 cm and between 10.0 cm and 25.0 cm (CI 95% 513; 732; *p* < 0.001). In relation to time, there were differences between 5.0 cm and 15.0 cm (CI 95% 0.330; 0.515; *p* < 0.001), 5.0 cm, and 25.0 cm (CI 95% 0.928; 1.345; *p* = 0.001), 10.0 cm and 15.0 cm (*p* < 0.05) and 15.0 cm and 25.0 cm (*p* < 0.05). No significant differences were observed between the muscles in electromyography, although the triceps showed the highest muscle activation values. Conclusions: The maximum isometric force, rate of force development, time, velocity, and dynamic time had lower values, especially in the initial and intermediate phases in the sticking region.

## 1. Introduction

The Powerlifting Paralympic (PP) is part of the sports calendar at the Paralympics, being a strength sport modality, adapted from conventional powerlifting, in which the athlete only performs the bench press. The bench press (BP) is a popular exercise, fairly simple to perform. In accordance with the rules and regulations set by the International Powerlifting Federation (IPF), the lift is performed lying on a bench with the head, shoulders, and buttocks in contact with the bench surface and the feet flat on the floor [1]. The lift begins with arms upright, the elbows stuck out. The barbell is then dropped to the chest or abdomen region until it is forced into straight arms and the elbows are locked out again at the end of the lift.

When performing the bench press at heavy load Madsen and McLaughlin [2] found that in the upward stroke of the bar, there is a point at which the lift slows down or even stops, before accelerating again. Particularly, during a raise at near-maximum loads (>80 percent of 1RM), there is a period where the upward barbell rotation decelerates or even ceases entirely for a brief period [3,4]. This period in which the driving force is less than gravity, leading to the slowdown of the barbell [4], is referred to as the “sticking period” or the “sticking region” [4,5]. This sticking area is also seen as the weakest link in the upward transition—a point at which the lift is likely to collapse [6]. The cause of this sticking region, however, remains unclear. Elliott et al. [5] found that the sticking area was not caused by a rise in the resulting moment arm on the shoulder or elbow or by a reduction in the muscle activity of the prime movers in that area. The leading interpretation model for the sticking zone is based on the assumption that the sticking area is a mechanically disadvantageous region for the development of force [7].

In this context, a possible interpretation of the sticking point is as electromyographic muscle activity (EMG). Few studies have analyzed muscle activity during the sticking point region [4,7]. Van den Tillaar and Ettema [4], concluded that a plausible explanation for the presence of the stuck phase is the reducing potentiation of the contractile elements during the upward step, along with the reduced activity of the pectoral and deltoid muscles during that time. However, in another study by Van den Tillaar, Saeterbakken, and Ettema [7], only the triceps increase in muscle activity during maximal lifting from the sticking to the post-sticking region. These studies did not investigate the effect of several different distances of the bar to the chest upon the muscle activation and sticking region. Investigating the different distances of the bar to the chest and muscle activation during the sticking region of these distances, which is a typical training method to increase maximal strength, would give information regarding whether the sticking region is influenced by the distance.

To our knowledge, no investigation has examined the sticking region and neuromuscular changes within and between the muscles in different distances of the bar to the chest in powerlifting athletes. In addition, possible changes in muscle activation within and between muscles during these distances can help us understand which muscles would be responsible for moving the barbell through the sticking region.

On the other hand, when evaluating the PP, where the legs are extended on the bench during the execution of the movement, situations such as the width of the grip on the bar [8] have been evaluated; however, a hypothesis that has been ignored is that a Paralympian’s transfer of force could be impaired making it difficult to maintain strength, power, and velocity and impairing the performance of neuromotor skills needed for lifting [1,9]. Thus, we raised the hypothesis, that due to the lifting position, with the lower limbs on the bench, and the physical deficiencies, which tend to compromise stabilization during the lifting, that the sticking point could be altered in these para-athletes. Therefore, the present study aimed to evaluate the strength indicators and the sticking point region in different distances of the bar to the chest in elite Powerlifting Paralympic athletes.

## 2. Materials and Methods

The athletes performed the tests of 1RM, velocity maximum (V_max_), maximum isometric force (MIF), rate of force development (RFD), time at MIF (Time), and surface electromyography (sEMG), distributed over five weeks as shown in Figure 1 (experimental design, weekly schedule of tests). At the beginning of each test day, the athletes performed a previous warm-up for the upper limbs, with three sets of 10 to 20 repetitions in approximately 20 min [10]. Then, a specific warm-up was performed on the bench press with 30% of the load for 1RM, where 10 slow repetitions and 10 fast repetitions were performed to start the procedure. These repetitions in a full range of motion served as a warm-up for the tests. During the test, the athletes received verbal encouragement to make the maximum effort. Each test followed its specific execution protocol, as previously described (Figure 2).

### 2.1. Participants

The subjects were 12 Powerlifting Paralympic athletes participating in the extension project of the Federal University of Sergipe, Sergipe, Brazil. All participants were Brazilian national level competitors, eligible for the competition [1] and ranked among the ten best in their respective categories. Among the athletes, two athletes were vice-champions of world cups, the third place in a world cup, two were champions, and two third-placed at a national level. In the observed deficiencies, we had four athletes presenting spinal cord injury due to accidents with an injury below the eighth thoracic vertebra; two with sequelae due to polio; four had lower limb malformation (arthrogryposis); and two had cerebral palsy. The characterization of the sample is shown in Table 1, where descriptive statistics based on the mean and standard deviation of each variable were used.

It is noteworthy that the sample size was defined a priori considering the η2p value of 0.766 found in the study by Ettema [7] (when comparing the variable strength at different distances in a male sample, in which the authors verified that the vertical distance of the bar was significant for the performance of the straight supine in healthy men); thus, a sample power of 0.98 (very strong) was estimated for a minimum sample of 12 subjects. For sample power analysis we used the open-source software G* Power (Version 3.0; Berlin, Germany) considering the *F* statistic with an alpha = 0.05 and a standard beta of 0.80. The analyses were repeated post hoc to expose the strength of the findings of the present study (details in the Section 3).

The athletes participated in the study voluntarily and signed a free and informed consent form, according to the Resolution 466/2012 of the National Research Ethics Commission-CONEP of the National Health Council, following the ethical principles expressed in the Helsinki Declaration (1964, reworded in 2013) of the World Medical Association. This study was approved by the Research Ethics Committee of the Federal University of Sergipe, CAAE: 2.637.882 (date of approval: 7 May 2018).

### 2.2. Instruments

Weighing of the athletes was performed on a digital type Michetti (Michetti, São Paulo, SP, Brazil) platform electronic scale, as this instrument facilitates the weighing of sitting athletes. The scale can hold a maximum weight of 300 kg and has dimensions of 1.50 × 1.50 m. For the bench press exercise, an official straight bench (Eleiko Sport AB, Halmstad, Sweden), approved by the International Paralympic Committee [1], with a total length of 210 cm was used. The IPC-approved powerlifting Olympic bar is serrated and has grooves in its material. They are 220 cm in total length, weighing 20 kg. The Olympic bar shall be marked with the narrowest and widest footprint, according to the International Paralympic Committee [1] official rules 2016–2017, ranging from 42 cm to 81 cm.

### 2.3. Load Determination

The 1RM test was performed, in which each subject started the attempts with a weight that he believed could lift only once using maximum effort. Weight increases were added until the maximum load that could be lifted once was reached. If the practitioner failed to perform a single repetition, 2.4% to 2.5% of the load used in the test was subtracted [12]. The subjects rested for 3–5 min between attempts. The test for determination of 1RM was made one week earlier, at least 48 h in advance so that it could determine the load percentage for the strength and power tests. The results are found in Table 1.

### 2.4. Determining Sticking Point Height

The sticking point occurs during the ascending (concentric) phase of the movement when performing the bench press exercise with a high, maximum, or sub-maximum load [4,5]. Thus, the bench press was performed following the standards of the IPC being performed with a downward motion until it touched the chest and then raising the bar until the elbows were extended. Stopping the bar at the chest tends to increase the occurrence of the sticking point region. It should be noted that isometric assessments in different bench press positions can be used to test the sticking region [4,5].

From the moment the value of 1RM was calculated, athletes rested 10 min [7,10] and then a load that was 2.0% to 2.5% of the load used in the test was added [12] until the subjects failed to raise the bar until the elbows were fully extended. The subjects rested for 3–5 min between attempts. From there, the distance from the bar to the chest where the athletes failed was determined. At this point, the sticking point was determined.

To confirm the height and also to evaluate the velocity in each phase, two phases (eccentric and concentric) were determined, and the concentric phase was divided into three periods, pre-sticking, sticking, and post-sticking. Between the eccentric and the concentric phase, there was a stop of the bar on the chest, which divided the movement and is mandatory in the Paralympic bench press mode [1].

### 2.5. Data Collection and Analysis

To evaluate the survey in the Bench Press Paralympic (vertical, horizontal, and displacement of the bar in the sagittal plane; a total distance of the bar path in the sagittal plane), dynamic time and the mean velocities of the bar were determined using a digitization software program (Kinovea 0.8.15, www.kinovea.org, accessed on 4 March 2020). The semi-automatic function was used to calculate the kinematic parameters, which was validated in practice [13,14]. For the collection of visual data, a digital video camera (frequency of 60 Hz and quality of 1080 *p*) was used. It was fixed at a height of 0.65 m and a distance of 1.25 m from the bar and aligned perpendicularly to the sagittal plane, for viewing the amplitude of movement. The movements were standardized, and the frames used were with the bar in the highest position (elbows extended) and the bar in the lowest position (against the chest). A 600% zoom was used to correctly identify the points (internal diameter of 2 mm) placed on the bar [13]. Two independent evaluators analyzed the videos, and when the difference was greater than 5%, a third evaluator examined the data.

### 2.6. Maximum Dynamic Velocity (V_max_) and Dynamic Time (DTime)

The athletes were evaluated during the competitive phase of the season and were familiar with the testing procedures due to their constant training and testing routines. V_Max_ was defined as 80% of 1RM, using only the weightless bar [15]. A Muscle Lab Encoder (Model PFMA 3010e Muscle Lab System; Ergotest, Langesund, Norway) was used and fixed to the end of the bar. A supine channel was adopted in order not to allow an incline of more than 2.0°, which allowed a dynamic evaluation of high reliability [16].

### 2.7. Maximum Isometric Strength (MIF), Rate of Force Development Rate (RFD), and Time to MIF

The measure of maximum isometric strength (MIF) (N), rate of force development rate (RFD) (N·s^−1^), and time to MIF (Time) (s) were determined by a Chronojump load cell (Chronojump, BoscoSystem, Madrid, Spain) with a capacity of 500 Kg, the output impedance of 350 ± 3 ohm, insulation resistance over 2000 cc, input impedance 365 ± 5 ohms, 24-bit, and 80 Hz digital–analog converter. The equipment was fixed to the adapted bench, using Spider HMS Simond carabiners (Sigmond Chamonix, Chamonix Mont-Blanc, France) with a breaking load of 21 KN, approved for climbing by the Union Internationaledes Associations d’Alpinisme (UIAA). A steel chain with a breaking load of 2300 kg was used to secure the load cell to the seat. The perpendicular distance between the load cell and the center of the joint was determined [17]. The maximum isometric force (MIF) was measured by the maximum strength generated by the muscles of the upper limbs in the adapted supine exercise [1]. The MIF was determined in Newton (N) conceived by the formula N = (M) × (C), where M = mass in Kg and C = 9.80665, measured between the load cell cable fixation point and the seat of the bench press. For the isometric test, the elbows were positioned at an angle close to 90°, angulation was verified with a device for measuring the amplitude of the angular movement, model FL6010 (Sanny^®^, São Bernardo do Campo, Brazil). Participants were instructed to perform a single maximum movement looking for elbow extension (as quickly as possible) for evaluation of the MIF; it was determined that the subjects had to maintain the maximum contraction for 5.0 s. The rate of force development (RFD), on the other hand, was determined using the force to time ratio until reaching the maximum force (RFD = ΔStrength/ΔTime) (N·s^−1^). The time to maximum force was determined as the time to reach the MIF. The MIF, RFD, and Time were evaluated with adaptations of the methodology of Van den Tillaar, Saeterbankken, and Ettema [7].

### 2.8. Surface Electromiography (sEMG)

The electromyographic signals were captured on the dominant side, using double electrodes, positioned parallel to the muscle fibers, 2 cm from the center at the point of greatest muscle area of the following muscles: brachial triceps (long head), anterior deltoid, and in the sternal and clavicular portions of the pectoralis major, on both sides of the body. The ground electrode was positioned over the olecranon. The skin area where the electrodes were placed was previously shaved and cleaned with alcohol. The electrodes (11 mm contact diameter and a 2 cm center-to-center distance) were placed along the presumed direction of the underlying muscle fiber according to the recommendations by SENIAM [18]. For data acquisition, one set was used with one repetition and a maximum load of 100% 1RM. The marker function was used to define the data intervals for each height in the sticking region.

The equipment used was an electromyographic MIOTEC^®^ with 8 channels (MIOTEC, Porto Alegre, RS, Brasil). Data were filtered (second-order Butterworth band-pass filter of 20–500 Hz; notch of 60 Hz). The signal amplitude was calculated through the mean square root (MSR), which was normalized by the percentage of the maximum voluntary isometric contraction (MVIC). MVIC acquisition occurred before the test was performed, and a lift was carried out that remained in an isometric state for 6.0 s. MVIC values were recorded by the equipment and used for normalization. The equipment program issues a report with the values after normalization that was used for analysis in this study, adapted from Golas et al. [19].

### 2.9. Statistics

Descriptive statistics were performed using measures of central tendency, mean (X) ± standard deviation (SD), and 95% confidence interval (95% CI). To verify the normality of the variables, the Shapiro–Wilk test was used given the sample size. The data for all variables analyzed were homogeneous and normally distributed. The one-way ANOVA test, with Bonferroni’s post hoc correction, was performed to evaluate the differences between phases and distance from the bar to the chest. All statistical analyses were performed using the computerized package Statistical Package for the Social Science (SPSS), version 22.0. The level of significance was set at *p* < 0.05. To check the effect size, (partial eta-squared: η2p) adopted values of low effect (≤0.05), medium effect (0.05 to 0.25), high effect (0.25 to 0.50), and very high effect (>0.50) [20].

## 3. Results

Figure 3 shows the velocity in movement (A) Conventional bench press and (B) Paralympic bench press.

In Table 2 are the results of velocity and dynamic time in each phase of the Paralympic Bench Press, in the eccentric and concentric phases (pre-sticking, sticking, and post-sticking). It is noteworthy that for the results related to velocity (m/s), we made a post hoc sample calculation considering the η2p value of 0.439; thus, a sample power of 0.63 (moderate) was indicated. We did the same procedure for the dynamic time (s) variable considering the η2p value of 0.769; thus, a sample power of 0.96 (very strong) was indicated. For sample power post hoc analysis, we used the open-source software G* Power (Version 3.0; Berlin, Germany) considering the *F* statistic (ANOVA-one way) with an alpha = 0.05 and a standard beta of 0.80.

In Figure 4, it was found that in relation to the MIF, the results showed differences between 5.0 cm (CI 95% 492; 802) and 15.0 cm (CI 95% 610; 895). In addition, the findings demonstrate singular statistical significance for the different distances analyzed. (a) There was a significant difference between 5.0 cm and 25.0 cm (95% CI 784; 1088; *p* = 0.010). (b) There was a significant difference between 10.0 cm and 5.0 cm (95% CI 527; 768; *p* < 0.001). (c) Differences were found between 10.0 cm and 15.0 cm (*p* = 0.012), and between 15.0 cm and 25.0 cm. (d) *p* = 0.012; (e) *p* = 0.007).

For the RDF, differences (*p* = 0.004) between 5.0 cm and 25.0 cm (95% CI 938; 1240) were evidenced. There was also a significant difference between 10.0 cm and 25.0 cm (CI 95% 513; 732; *p* < 0.001). There were no significant differences between the other heights of the bar to the chest.

In relation to velocity, differences (*p* < 0.05) were found between 5.0 cm (95% CI 0.614; 0.724) and 10.0 cm (95% CI 0.140; 0.256). Thus, the results pointed out some peculiarities in relation to the analyzed distances: (a) Differences were found between 5.0 cm and 15.0 cm (CI 95% 0.330; 0.515; *p* < 0.001). (b) Differences were found between 5.0 cm and 25.0 cm (95% CI 0.928; 1.345; *p* = 0.001). Additionally, differences were found between 10.0 cm and 15.0 cm. (c) *p* = 0.002; (d) *p* = 0.040. Differences were found between 10.0 cm and 25.0 cm (e) *p* < 0.001; and differences were found between 15.0 cm and 25.0 cm (f) *p* = 0.001.

The surface electromyography (Figure 5) indicated greater activation of the brachial triceps in comparison with the other muscles, mainly at 10.0 cm and 15.0 cm distances. In addition, no significant differences were observed between the studied muscles and between the different distances from the bar to the chest.

## 4. Discussion

The objective of the present study was to evaluate the strength indicators in the region of the sticking point in Paralympic weightlifting athletes. In this sense, the main findings were: (i) There was a significant difference for the maximum isometric strength between 5.0 cm and 15 cm, as well as between 5.0 cm and 10.0 cm. (ii) For the rate of force development, differences were found between 5.0 cm and 25.0 cm and between 10.0 cm and 25.0 cm. (iii) For velocity, differences between 5.0 cm and 15.0 cm, between 5.0 cm and 25.0 cm, and between 10.0 cm and 15.0 cm were evidenced. (iv) No significant differences were observed between muscles on electromyography, although the triceps showed the highest values of muscle activation. (v) The maximum isometric force, the rate of force development, and the velocity presented lower values, especially in the initial and intermediate phases in the sticking point region.

The maximum isometric strength showed significant differences between 5.0 and 15.0 cm (*p* = 0.001), between 5.0 and 25.0 cm (*p* < 0.001), between 10.0 and 15.0 cm (*p* = 0.012), and between 15.0 and 25.0 cm (*p* = 0.007). This shows an increase in strength between the initial and final distances—mainly after the sticking region. The study by Van den Tillaar, Saeterbankken, and Ettema [7] investigated the effects of performing the bench press in an isometric way at different moments from the chest bar (0 to 31 cm) and dynamically using 1RM. The study showed a decrease in strength between 4 cm and 13 cm away from the bar on the sternum bone. It also indicated that the FIM increased gradually between the sticking region (4.0 to 13.0 cm), surpassing the force generated by the regular bench press from the end of the sticking point and making it significant from 22.0 cm away from the chest.

Regarding the rate of force development (RFD), our study found significant differences between 5.0 and 25.0 cm (*p* = 0.004) and between 10.0 and 25.0 cm (*p* < 0.001). There were no significant differences between the other heights of the bar to the chest. Corroborating our findings, Drinkwater et al. [24] mention that during the concentric movement of the bench press, there is an initial high-power impulse after contact of the bar with the chest, which tends to be followed immediately by a moment of low power, called sticking point. Thus, a decrease in power and the rate of force development in high-intensity jobs tends to lead to movement failure. If we consider that the RFD would be the capacity to generate strength in less time, this capacity would be a good parameter with which to measure the neuromuscular efficiency of the athletes [25]. The results of our study demonstrate a greater RDF in further distances in the sticking region, which may be a justification for the failure in the region.

In this sense, the sticking point would be more associated with the perspective of the manifestation of strength and conditioning for preventing injuries, in addition to progress in strength adaptations for athletes [26]. Thus, considering that the Paralympic athletes are ranked among the strongest in Brazil, this would justify the increase in RDF to the points 5.0 cm, 10.0 cm, 15.0 cm, and mainly post-sticking point of 25.0 cm.

Regarding velocity, our findings point to significant differences between 5.0 and 10.0 cm (*p* < 0.001); between 5.0 and 15.0 cm (*p* = 0.001); and between 5.0 and 25.0 cm (*p* = 0.002). There were also differences between 10.0 and 15.0 cm (*p* = 0.040) and between 10.0 and 25.0 cm (*p* < 0.001). There was also a difference between 15.0 and 25.0 cm (*p* = 0.001).

In the same direction Gomo and Van Der Tillaar [27] found that the peak velocity and the local minimum velocity occurred with more closed footprints, which was not the target of our study. However, our athletes by competition regulations cannot make wider footprints than 81.0 cm (IPC, 2018), which would be relatively closed footprints. The minimum velocity of the sticking region would occur when the upward movement of the bar decelerated or even stopped completely for a short time during the lift [4]. Our findings indicate that at 5.0 cm we have a high velocity (0.699 m/s), and in intermediate sticking at 10.0 cm velocity tends to fall (0.198 m/s) and then increases by 15 cm (0.423 m/s) and 25.0 cm (1.137 m/s), showing changes in velocity at the various points in the sticking region.

In this sense, this study analyzed the muscle activation (electromyography) of the main muscles involved in the kinetics of this exercise in the sticking region. The results showed greater activation of the triceps compared with the other muscles analyzed. Corroborating this result, other studies [4,28] also indicate greater activation of the triceps, compared with other muscles involved in the bench press. Thus, it is possible to consider that the muscular activation of the triceps contributes to overcoming the sticking region during the lifting of the bar in the concentric phase of the bench press.

However, despite the relevance of the results, this study has some limitations. The sample consisted of national athletes with different disabilities eligible for the modality. In this sense, the findings are for practitioners of Paralympic powerlifting, not evaluating possible differences that could happen in the various disabilities eligible for the sport, since the PP has a unique class, where athletes are not separated by type of disability like in other sports. However, the current findings are still relevant for coaches and researchers for a greater understanding of the sticking point and its effects on sports performance.

## 5. Conclusions

Thus, it is possible to conclude that in the sticking region, the maximum isometric force, the rate of force development, time, and velocity tend to be impaired to the beginning and after the sticking region and also that the muscular activation of the triceps stands out in all intervals from the sticking point.

The determination of the sticking point becomes important so that coaches can focus on this region, and thus having appropriate and effective training for the point at which the failure normally occurs increases the possibility of improving powerlifting results.

## Figures and Tables

**Figure 1 jfmk-06-00043-f001:**
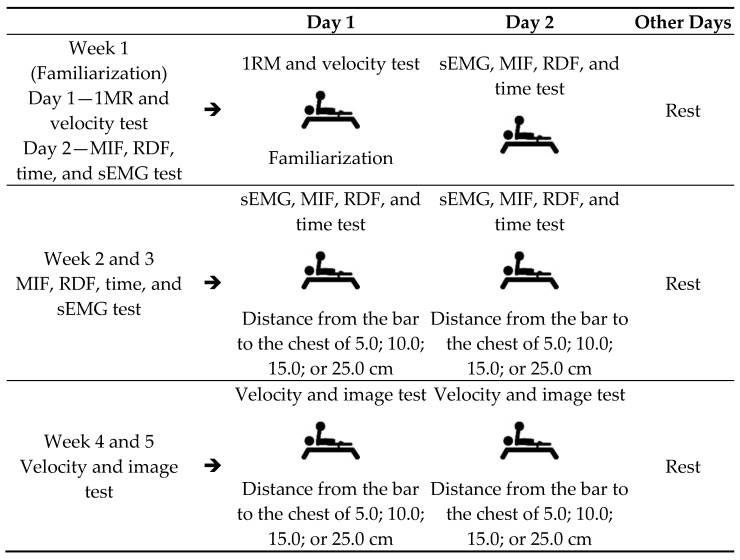
Experimental design: Weekly test schedule. Legend: 1RM: 1 Repetition Maximum; SEMG: surface electromyography; MIF: Maximum Isocmetric Force; RFD: Rate of Force Development; Time: Time to MIF.

**Figure 2 jfmk-06-00043-f002:**
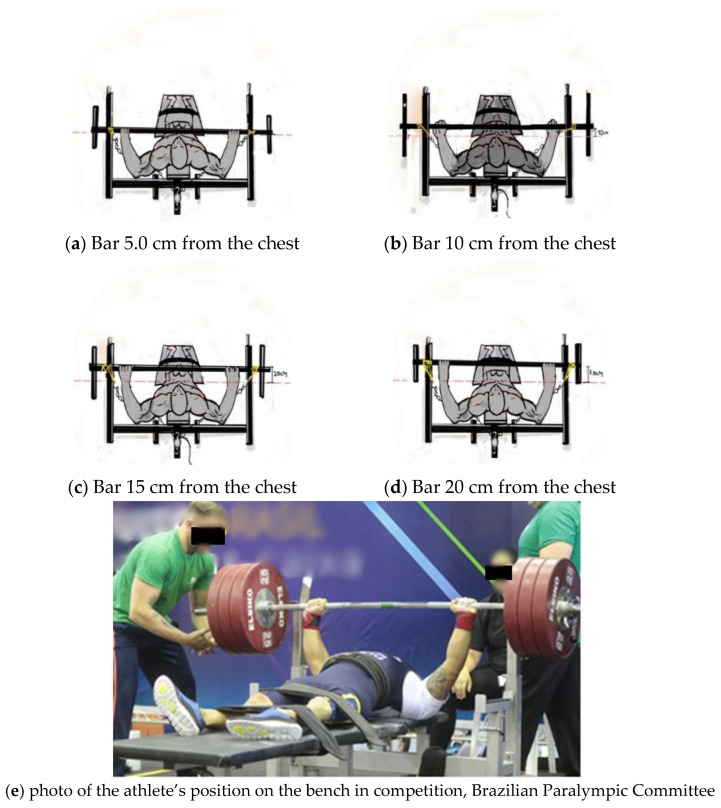
Schematic drawing with the different heights (**a**–**d**) of the bar to the chest and (**e**) photo of the athlete’s position on the bench in competition.

**Figure 3 jfmk-06-00043-f003:**
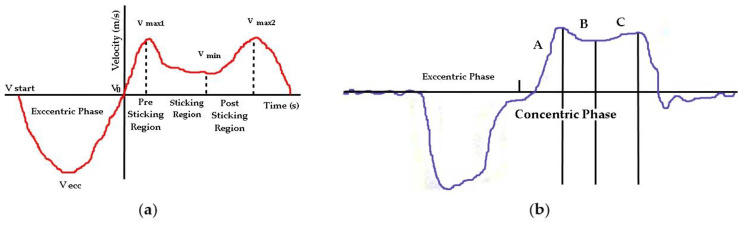
(**a**) Representative velocity–time curve during (sub) maximal bench press lifts (>85% of 1RM) in conventional Bench Press [21,22,23] and (**b**) representative velocity–time curve during (sub) maximal bench press lifts (95% of 1RM) in Paralympic Power Lifting with the different events and regions. Legend: V: velocity, V_0_: at the beginning of the concentric phase, V ecc: Velocity in the eccentric phase, Vmax1: maximum velocity at point 1, Vmax2: maximum velocity at point 2, Vmin: Minimum velocity, (m/s): Meters per second, (s): Second; A: pre-sticking period, B: sticking period; C: post-sticking period.

**Figure 4 jfmk-06-00043-f004:**
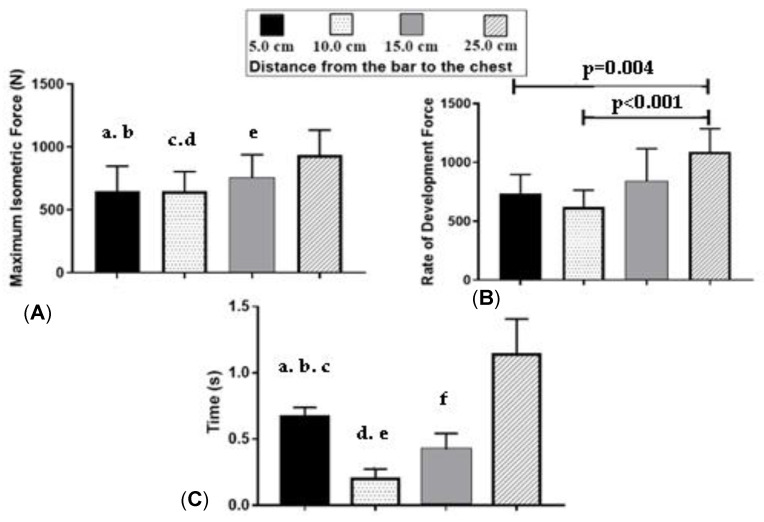
(**A**) Maximum isometric force (MIF), (**B**) rate of force development (RFD), and in relation to (**C**) time, with 5.0; 10.0; 15.0; and 25. cm of the distance from the bar to the chest.

**Figure 5 jfmk-06-00043-f005:**
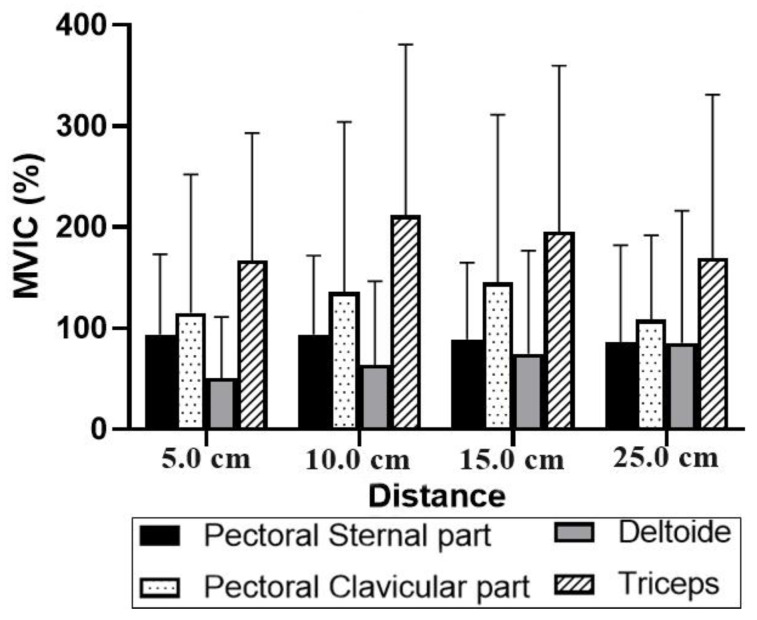
Electromyography of the pectoral, sternal and clavicular parts, deltoid and triceps at different distances from the chest. Legend: MVCI: maximal voluntary isometric contraction (%).

**Table 1 jfmk-06-00043-t001:** Characterization of subjects.

Variables	Values
Age (years)	26.56 ± 5.55
Body Weight (Kg)	77.89 ± 24.60
Experience (years)	3.47 ± 0.27
Footprint Width (cm)	59.44 ± 17.77
1RM Bench Press Test (Kg)	135.56 ± 30.15 *
Sticking Point (cm)	10.01 ± 3.31 **
1RM/Body Weight	1.81 ± 0.35 ***

* All athletes with loads that keep them in the top 10 of their national categories. ** In the sticking region, the minimum value was 5.0 cm, and the maximum was 15.00 cm. *** Bench press values above 1.4 would be considered elite athletes, according to Ball and Wedman [11].

**Table 2 jfmk-06-00043-t002:** Mean ± SD values and 95% confidence interval (CI 95%) of velocity and time to achieve eccentric, pre-sticking, sticking, and post-sticking in Paralympic Powerlifting.

	Eccentric X ± DP (IC 95%)	Pre-Sticking X ± DP (IC 95%)	Sticking X ± DP (IC 95%)	Post-Sticking X ± DP (IC 95%)	*p*	η2p
Velocity (m/s)	1.62 ± 0.26 (1.42; 1.83)	1.98 ± 0.32 * (1.73; 2.23)	1.30 ± 0.43 * (0.97; 1.64)	1.86 ± 0.37 (1.58; 2.15)	* 0.039	0.439 #
Dynamic Time (s)	1.81 ± 0.50 a (1.43; 2.19)	0.40 ± 0.16 a,b (0.28; 0.52)	0.97 ± 0.37 a,b (0.68; 1.25)	0.85 ± 0.31 a (0.62; 1.09)	a < 0.001 b = 0.021	0.769 ##

* *p* < 0.05 (ANOVA, one-way); η2p = partial eta-squared; #, high effect (0.25 to 0.50); ##, very high effect (>0.50).

## Data Availability

The data that support this study can be obtained from the address www.ufs.br/GPEPS, accessed on 20 February 2021.

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
