# Peer review of "Evaluation of Strength and Muscle Activation Indicators in Sticking Point Region of National-Level Paralympic Powerlifting Athletes"

_jfmk, 2021, doi:10.3390/jfmk6020043_

Round 1

Reviewer 1 Report

The authors articulate the issue clearly in terms of the practicality of the underlying question.

The authors should make a case why the sample and population being studied represents a particularly useful way of starting this line of research. The authors have a group of experienced athletes and could make more of this in their work. I would refer to them as participants rather than subjects. I would explain how their varying levels of disability could have impacted upon the ability to perform this test.

The data analysis section needed more depth and explain how the underlying hypothesis would be examined. This is particularly relevant for the section on the statistical analysis where sample size seems extremely small and this does not seem to be considered in their approach. In the results I'm looking to see the degrees of freedom so that I can see how many data points were used.

Having not being able to follow the results clearly enough from the perspective of being able to interpret what the statistical tests are showing in relation to how many people or data points are being used I found it hard to then follow this work through into the discussion. The data analysis section seems critical and along with this it would help if the authors presented hypothesis or expectations of what the results are likely to say.

The authors should provide the practical application of the research having set the work up in that way this would be a suitable conclusion.

Author Response

Reviewer 1

The authors articulate the issue clearly in terms of the practicality of the underlying question.

The authors should make a case why the sample and population being studied represents a particularly useful way of starting this line of research. The authors have a group of experienced athletes and could make more of this in their work. I would refer to them as participants rather than subjects. I would explain how their varying levels of disability could have impacted upon the ability to perform this test.

Answer: A sentence was inserted on page 12 to better exemplify the sample.

The data analysis section needed more depth and explain how the underlying hypothesis would be examined. This is particularly relevant for the section on the statistical analysis where sample size seems extremely small and this does not seem to be considered in their approach. In the results I'm looking to see the degrees of freedom so that I can see how many data points were used.

Answer: It has been adjusted for better understanding (abstract and pages 7-8).

Having not being able to follow the results clearly enough from the perspective of being able to interpret what the statistical tests are showing in relation to how many people or data points are being used I found it hard to then follow this work through into the discussion. The data analysis section seems critical and along with this it would help if the authors presented hypothesis or expectations of what the results are likely to say.

Answer: It has been adjusted for better understanding (abstract and pages 7-8).

The authors should provide the practical application of the research having set the work up in that way this would be a suitable conclusion.

Answer: The conclusion section has been improved for a better understanding.

Reviewer 2 Report

The aim of this study was to evaluate the strength indicators and the sticking point region in different distances of the bar to the chest in elite Powerlifting Paralympic athletes.

The work has been well conducted, however some suggestions are proposed with the intention of improving some aspects.

1. Although in line 156 it is specified that the movement to determine the sticking point height is concentric, it is suggested that the procedure to determine the 1 RM should also be specified.

2. Line 99.- Briefly specify the type of repetitions performed.

3. Review the line endings (for example, line 182, 215 ...).

4. Lines 271-272.- The phrase is not understood. What has been found?

5. Check the use of parentheses in the results section (double and misused parentheses appear).

6. Line 306. Maintain the original Sticking Point concept.

7. It is recommended to include a paragraph or section on the possible limitations of the study.

8. It is recommended to include, in the conclusions section, the implications of the findings provided.

Author Response

Reviewer 2

The aim of this study was to evaluate the strength indicators and the sticking point region in different distances of the bar to the chest in elite Powerlifting Paralympic athletes.

The work has been well conducted; however, some suggestions are proposed with the intention of improving some aspects.

  1. Although in line 156 it is specified that the movement to determine the sticking point height is concentric, it is suggested that the procedure to determine the 1 RM should also be specified.

Answer: It has been adjusted for a better understanding (page 5).

  1. Line 99.- Briefly specify the type of repetitions performed.

Answer: It has been adjusted for a better understanding (page 2-3).

  1. Review the line endings (for example, line 182, 215 ...).

Answer: It has been adjusted for a better understanding

  1. Lines 271-272.- The phrase is not understood. What has been found?

Answer: It has been adjusted for a better understanding

  1. Check the use of parentheses in the results section (double and misused parentheses appear).

Answer: It has been adjusted for a better understanding

  1. Line 306. Maintain the original Sticking Point concept.

Answer: It has been adjusted for a better understanding

  1. It is recommended to include a paragraph or section on the possible limitations of the study.

Answer: A paragraph was written about the limitations of the study.

  1. It is recommended to include, in the conclusions section, the implications of the findings provided.

Answer: It has been adjusted for a better understanding

Round 2

Reviewer 1 Report

The authors have revised the paper and it is much better.

I would ask for the authors to expand and explain the point made in this paragraph. Practitioners are coaches in my mind.  They should expand and explain.  

'In this sense, the findings are for practitioners of the sport in general, however, they do not offer an insight into the sticking point in each type of disability. However, the current  findings are still relevant for coaches and researchers for a greater understanding of the sticking point and its effects on sports performance'

Change subjects to participants for consistency throughout.

Put the spellchecker on English in word and re-run. You will find some spelling mistakes. 

Author Response

Reviewer 1

Comments and Suggestions for Authors

The authors have revised the paper and it is much better.

I would ask for the authors to expand and explain the point made in this paragraph. Practitioners are coaches in my mind. They should expand and explain.

'In this sense, the findings are for practitioners of the sport in general, however, they do not offer an insight into the sticking point in each type of disability. However, the current findings are still relevant for coaches and researchers for a greater understanding of the sticking point and its effects on sports performance'

Change subjects to participants for consistency throughout.

Put the spellchecker on English in word and re-run. You will find some spelling mistakes. 

Answer: Thank you, and the adjustments were made as requested